# Haldane topological spin-1 chains in a planar metal-organic framework

Pagnareach Tin [1], Michael J. Jenkins[1], Jie Xing[2], Nils Caci [3], Zheng Gai [4], Rongyin Jin [2], Stefan Wessel [3], J. Krzystek[5], Cheng Li [6], Luke L. Daemen[6], Yongqiang Cheng [6] & Zi-Ling Xue [1] ✉

Haldane topological materials contain unique antiferromagnetic chains with symmetry-protected energy gaps. Such materials have potential applications in spintronics and future quantum computers. Haldane topological solids typically consist of spin-1 chains embedded in extended three-dimensional (3D) crystal structures. Here, we demonstrate that $[Ni(\mu-4,4'-bipyridine)(\mu-oxalate)]_n$ (NiBO) instead adopts a two-dimensional (2D) metal-organic framework (MOF) structure of $Ni^{2+}$ spin-1 chains weakly linked by 4,4'-bipyridine. NiBO exhibits Haldane topological properties with a gap between the singlet ground state and the triplet excited state. The latter is split by weak axial and rhombic anisotropies. Several experimental probes, including single-crystal X-ray diffraction, variable-temperature powder neutron diffraction (VT-PND), VT inelastic neutron scattering (VT-INS), DC susceptibility and specific heat measurements, high-field electron spin resonance, and unbiased quantum Monte Carlo simulations, provide a detailed, comprehensive characterization of NiBO. Vibrational (also known as phonon) properties of NiBO have been probed by INS and density-functional theory (DFT) calculations, indicating the absence of phonons near magnetic excitations in NiBO, suppressing spin-phonon coupling. The work here demonstrates that NiBO is indeed a rare 2D-MOF Haldane topological material.

Haldane indicated in 1983 that antiferromagnetic (AF) Heisenberg chains of integer spin exhibit a quantum-disordered ground state and a finite energy gap, whereas the half-integer spin chains are gapless, based on effective-field theory and topological considerations[1,2]. Fig. 1 shows the energy diagram in a Haldane spin-1 chain system, including the anisotropic (zero-field splitting or ZFS) and Zeeman effect. The Haldane phase of the AF spin-1 chain is regarded as the most fundamental example of symmetry-protected topological states[3–8]. Following Haldane's work, Affleck, Kennedy, Lieb, and Tasaki (AKLT) identified an extended spin-1 model Hamiltonian describing the

ground state of the Haldane phase with one unpaired electron at each end of the chain ($S = 1/2$)[9–13], as visualized in Fig. 2a. The two electrons at the ends of the chain lead to singlet or triplet edge states. The topological Haldane phase is unique due to the existence of the finite energy gap that is symmetry-protected[4–7]. As long as the set of symmetries is preserved, the Haldane phase is stable and can withstand external perturbations[6,8]. The Haldane materials are in contrast to typical antiferromagnetic materials such as NiO with long-range magnetic ordering in Fig. 2b[14]. In NiO, a magnetic unit cell forms in addition to its crystal unit cell, leading to the observation of Bragg reflections

[1]Department of Chemistry, University of Tennessee, Knoxville, TN 37996, USA. [2]Center for Experimental Nanoscale Physics, Department of Physics and Astronomy, University of South Carolina, Columbia, SC 29208, USA. [3]Institut für Theoretische Festkörperphysik, RWTH Aachen University, 52056 Aachen, Germany. [4]Center for Nanophase Materials Sciences, Oak Ridge National Laboratory, Oak Ridge, TN 37830, USA. [5]National High Magnetic Field Laboratory, Florida State University, Tallahassee, FL 32310, USA. [6]Neutron Scattering Division, Oak Ridge National Laboratory, Oak Ridge, TN 37830, USA. ✉e-mail: xue@utk.edu

from the magnetic unit cell together with peaks from the nuclear diffraction of its crystal unit cell[14].

Properties of 1D Haldane spin-1 materials have been probed experimentally using, e.g., $Ni^{2+}$-based chains bridged by $NO_2^-$, $N_3^-$, $ox^{2-}$, or $HF_2^-$ ligands[3], such as $[Ni(en)_2(\mu\text{-}NO_2)](ClO_4)$ (NENP; en = ethylenediamine)[15–20], $[Ni(1,3\text{-propanediamine})_2(\mu\text{-}NO_2)](ClO_4)$ (NINO)[20,21], $[Ni(en)_2(\mu\text{-}NO_2)](BF_4)$ (NENB)[22–24], $[Ni(dmpn)_2(\mu\text{-}N_3)](ClO_4)$ (NDMAZ; dmpn = 1,3-diamino-2,2-dimethylpropane)[25], $[Ni(3,2,3\text{-tet})(\mu\text{-}N_3)](ClO_4)$ [232-tet = bis(2-aminoethyl)-l,3-propanediamine][26], $[Ni(2,2'\text{-bpy})(\mu\text{-}ox)]_n$ (bpy = bipyridine)[27], $[Ni(benzimidazole)_2(\mu\text{-}ox)]_n$[28], and $[NiI_2(3,5\text{-lut})_4]_n$ (3,5-lut = 3,5-lutidine)[29]. In the 1D chains with mono-anionic $\mu\text{-}NO_2^-$ or $\mu\text{-}N_3^-$ ligands bridging $Ni^{2+}$ ions such as NENP or

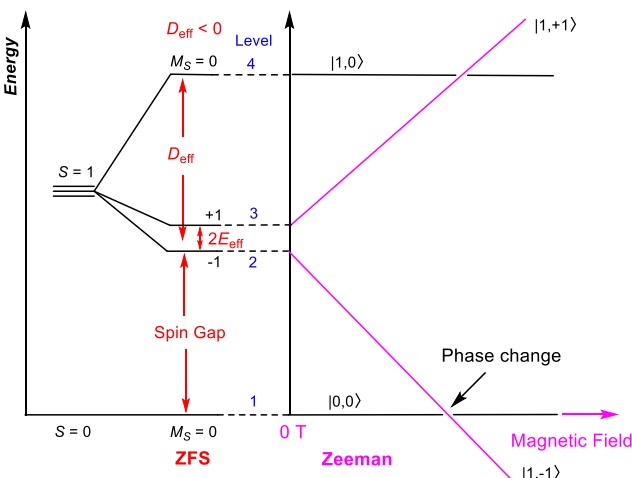

**Fig. 1 | Energy diagram of a Haldane spin-1 chain system and the Zeeman effect on the magnetic states.** Here, $D_{eff}$ and $E_{eff}$ are effective axial and rhombic (or transverse) ZFS parameters for the lowest spin excitations, respectively. The right panel illustrates the Zeeman effect on the states, leading to a magnetic phase transition at about 9 T for NiBO. The labels "+1" and "−1" represent $M_S$ = +1 (or $|1,+1\rangle$) and −1 (or $|1,-1\rangle$) states, respectively, of the excited triplet state. The energy splitting diagram is based on the $z$ direction [i.e., along the Ni(ox) chains as shown in Fig. 2] aligned parallel to the magnetic field.

NENB, the Haldane chains carry + charges, requiring anions $ClO_4^-$ or $BF_4^-$ to balance the charges. In $[Ni(HF_2)(pyz)_2]SbF_6$[30], the mono-anionic $HF_2^-$ ligands also form cationic 1D linear chains along the $c$ axis in the coordination polymer, using the $SbF_6^-$ anions to balance the + charges. In $[Ni(2,2'\text{-bpy})(\mu\text{-}ox)]_n$[27] $[Ni(benzimidazole)_2(\mu\text{-}ox)]_n$[28], the N-ligands, $2,2'$-bpy or benzimidazole, are *cis*-coordinated to $Ni^{2+}$ ions, forming zig-zag structures.

Haldane spin-1 chains based on MOFs with oxalate-based ligands, $[Ni(\mu\text{-bpa})(\mu\text{-ox})]_n$ (bpa = 1,2-bis(4-pyridyl)ethane)[31], $[Ni(\mu\text{-bpe})(\mu\text{-ox})]_n$ (bpe = 1,2-di(4-pyridyl)ethylene)[31], $[Ni(\mu\text{-en})(\mu\text{-ox})]_n$[32], and $[Ni(\mu\text{-pip})(\mu\text{-ox})]_n$ (pip = piperazine)[33], have been reported to show the presence of a finite spin gap through DC magnetic susceptibility measurements. Two features distinguish these Haldane spin-1 chains: (1) The oxalate ligands ($ox^{2-}$) in the spin-1 chains carries −2 charge balancing the +2 charge of the $Ni^{2+}$ ions. Thus, these spin-1 chains based on oxalate are neutral in contrast to the positively charged spin-1 chains in, e.g., NENP and NDMAZ. (2) The MOF-based compounds exhibit stacked 2D planar structures with Haldane 1D chains linked by N-containing ligands. In contrast, cationic Haldane chains containing anions in NENP and NDMAZ make the solids non-planar. The 2D-MOFs may be exfoliated by overcoming the relatively weak Van der Waals force between the layers. Exfoliation of the 2D-MOFs can yield 2D nanosheets with interesting low-dimensional properties. Several exfoliation methods have been established to be effective in synthesizing the 2D-MOF nanosheets[34–36]. From DC magnetic susceptibility data, García-Couceiro and coworkers found that the spin gap = 7.32 and 10.29 $cm^{-1}$ for $[Ni(\mu\text{-bpa})(\mu\text{-ox})]_n$ and $[Ni(\mu\text{-bpe})(\mu\text{-ox})]_n$, respectively[31]. Similarly, Keene and coworkers obtained spin gap = 11.5 $cm^{-1}$ for $[Ni(\mu\text{-en})(\mu\text{-ox})]_n$ from DC susceptibility data[32]. The spin gaps are comparable to those of other spin-1 chain systems such as NENP and NENB[15,22]. The results show that 2D-MOFs containing inter-chain ligands that form planar structures have the potential to exhibit Haldane spin-chain physics, as long as the inter-chain magnetic couplings is negligible.

Here, we report that $[Ni(\mu\text{-}4,4'\text{-bpy})(\mu\text{-ox})]_n$ (NiBO)[37] is a Haldane spin-1 chain material. It has been comprehensively probed by a variety of techniques, including variable-temperature powder neutron diffraction (PND), magnetization measurements (DC magnetic susceptibility at different fields), high-field ESR (HFESR), inelastic neutron scattering (INS), and specific-heat measurements. In addition, we

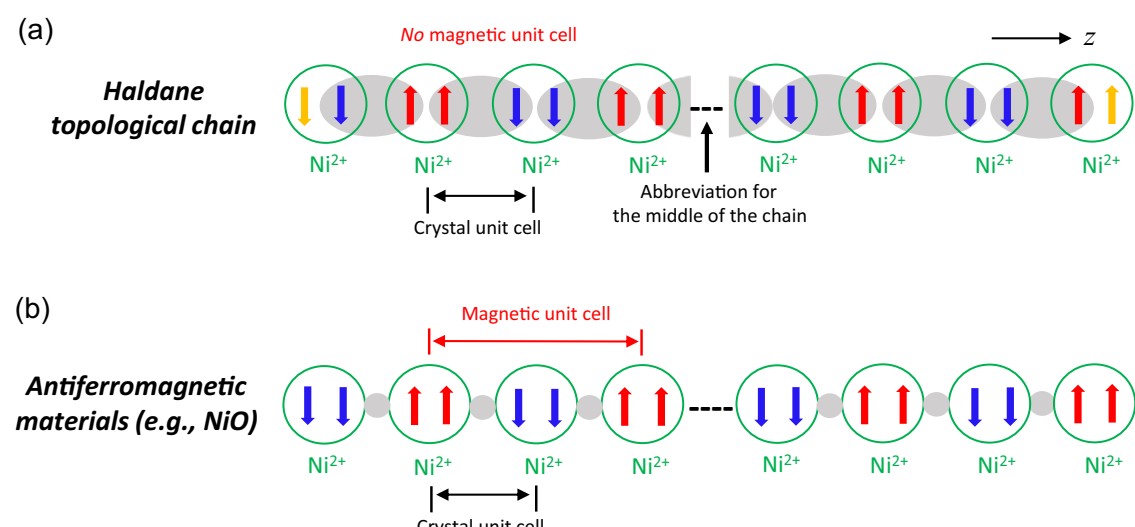

**Fig. 2 | Differences between a Haldane topological chain and an antiferromagnetic material. a** AKLT model representation of the $Ni^{2+}$ spin-1 Haldane chain showing the antiferromagnetic coupling (gray oval) between two unpaired electrons (red and blue arrows) in the middle of the chain, leaving unpaired electrons (orange arrows) at ends of the chain. These two unpaired electrons may have antiparallel (shown) or parallel spins, giving singlet and triplet states, respectively, in Fig. 1. There is no magnetic unit cell in the Haldane system. **b** Schematic of the antiferromagnetic structure in NiO below its Néel temperature. The magnetic unit cell has twice the linear dimension of the crystal unit cell, as revealed by PND[14]. In the crystal unit cell, $Ni^{2+}$ ions form a face-centered cubic cell with ferromagnetically coupled sheets that are anti-parallel with adjacent sheets. $O^{2-}$ ions are shown as small gray cycles.

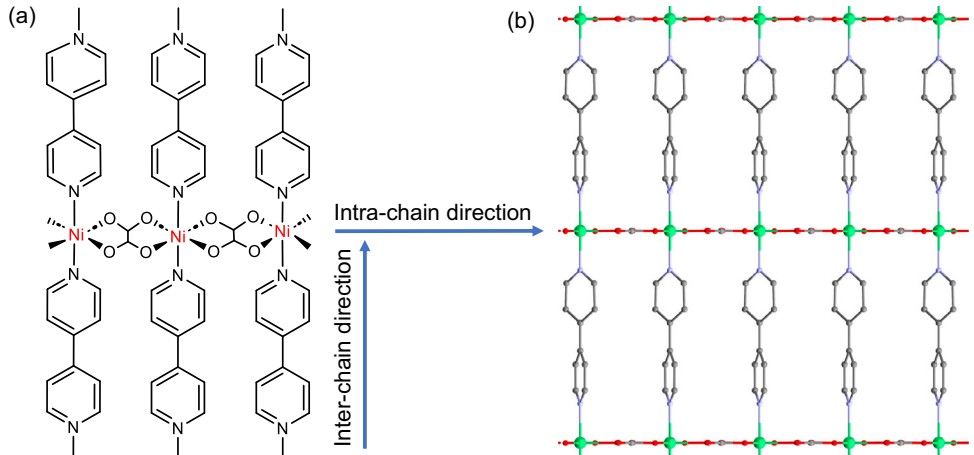

**Fig. 3 | Schematic of NiBO and the crystal structure of NiBO-$d_8$. a** 2D schematic of NiBO showing the bonding around three $Ni^{2+}$ ions within the chain. Blue arrows show intra- and inter-chain directions. **b** Structure of NiBO-$d_8$ viewed down the crystallographic $a$-axis. Green: Ni; Red: O; Purple: N; Gray: C. Selected bond lengths and angles: Ni-O = 2.0488(10)-2.0533(9) Å, Ni-N = 2.092(5) Å. *cis*-ligand angles: O-Ni-O 82.53(4)-97.47(4)°, O-Ni-N 89.68(11)-90.32(11)°; *trans*-ligand angles: O-Ni-O 179.4(2)-179.5(2)°, N-Ni-N 180.0°. The CIF (Crystallographic Information Framework) file of the crystal structure, which has been deposited in the Cambridge Structural Database (CCDC No. 2278974), is provided as a Source Data file.

performed quantum Monte Carlo (QMC) simulations of the spin-1 chains realized in NiBO, which show excellent agreement with the experimental DC magnetic susceptibility data. VT-PND data exhibit no long-range antiferromagnetic ordering down to 1.7 K, further supporting that NiBO is a quantum-disordered spin-1 chain system. The spin gap extracted from the DC magnetic susceptibility measurements at 0.1 T agrees remarkably well with QMC simulations. The QMC results also identify magnetic parameters such as $D$, $E$, $J$, and $g$-factors ($g$) for NiBO, as detailed below. Additionally, both the magnetic susceptibility data and specific-heat data reveal a field-induced phase transition in NiBO for magnetic field beyond 9 T and temperatures below 5 K. HFESR was used to observe the effective anisotropy ($D_{eff}$) in ZFS through the transition within the triplet state. Furthermore, these magnetization measurements are well in accord with QMC simulations. X-ray single-crystal diffraction of deuterated [Ni($\mu$−4,4'-bpy-$d_8$)($\mu$·ox)]$_n$ (NiBO-$d_8$) at 100 K revealed a non-planar structure of the 4,4'-bpy-$d_8$ ligand, which reduces inter-chain magnetic interactions. Inelastic neutron scattering (INS) studies provide the spin gaps between the singlet ground and the triplet excited state as well as the phonon properties of NiBO, which are supported by density-functional theory (DFT) calculations using the Vienna Ab initio Simulation Package (VASP)[38].

## Results

### Single-crystal X-ray diffraction

NiBO and its crystal structure at 295 K were reported earlier[37]. In the current work, single-crystal X-ray diffraction on NiBO-$d_8$ was performed at 100(2) K, providing an in-depth analysis of its structure to compare with that reported at 295 K (Supplementary Table 1)[37]. Each $Ni^{2+}$ ion is coordinated with two oxalate ligands in a plane as intra-chain bridging ligands forming the spin-1 chains, while two 4,4'-bpy-$d_8$ ligands coordinate in the $z$-direction as inter-chain bridging ligands, resulting in an overall 2D framework. There is only one unique NiBO-$d_8$ unit (one oxalate, one 4,4'-bpy-$d_8$ ligand, and one $Ni^{2+}$ ion) in the crystal structure of the MOF. Fig. 3 and Supplementary Fig. 5 illustrate the inter- and intra-chain directions of the 1D spin chains in NiBO as well as a view down the $a$-axis. At 100 K, we have found that the two pyridine rings in the 4,4'-bpy-$d_8$ ligands are not co-planar, likely due to the free rotational movement of the two rings along the C-C bond in thermal environment. The resulting effect could likely reduce magnetic interactions between neighboring spin chains due to limiting π-conjugations in the 4,4'-bpy-$d_8$ ligand, a

desirable property for a Haldane material. Although the co-planar structure of the two rings in the 4,4'-bpy-$d_8$ ligand provides a larger π-conjugation, rotations of the two pyridine rings by thermal energy may have led to the non-co-planar structure. Further discussions of the single-crystal structures are provided in the Supplementary Note 1.

### Variable-temperature powder neutron diffraction

PND was conducted at 1.7, 10, 20, 100, and 200 K to study the magnetic properties of NiBO and any potential structural phase transition at zero-field. As shown in Fig. 2, in the Haldane phase, there is no magnetic ordering. Thus, only nuclear diffraction from the crystal unit cell is expected, whether the sample is in the paramagnetic or the low-temperature Haldane phase. In other words, no additional PND peaks are expected when the sample is cooled. In comparison, for antiferromagnetic NiO with long-range magnetic ordering (Fig. 2b)[14], diffractions from both the magnetic unit cell and the crystal unit cell are observed[14].

No change was observed in the PND pattern between 1.7 and 200 K for a powder sample of NiBO. Fig. 4a shows a comparison between the 1.7 and 100 K diffraction patterns, indicating *no* long-range antiferromagnetic correlations, as extra magnetic Bragg reflections were not observed at low temperature. The absence of such changes in NiBO indicates that the nature of its antiferromagnetic interaction is not of a long-range ordered state but is consistent with the Haldane phase in Fig. 2. Furthermore, the Rietveld refinement of the PND pattern at 1.80 K reveals no long-range ordering or magnetic phase transition in NiBO at zero-field (Fig. 4b), consistent with results obtained from both the magnetization and specific heat discussed below.

### Magnetic susceptibility (DC)

The temperature dependence of the DC magnetic susceptibility on a powder sample of NiBO at 0.1 T applied external magnetic field is plotted in Fig. 5a. Upon lowering the temperature from 300 K, the magnetic susceptibility (χ) first increases and exhibits a broad maximum around 52 K, below which it decreases down to near zero magnetic susceptibility at 2 K. In the 200−300 K range, the data can be well fitted to the Curie-Weiss law[39]:

$$\chi = C/(T - \theta_{CW}) \quad (1)$$

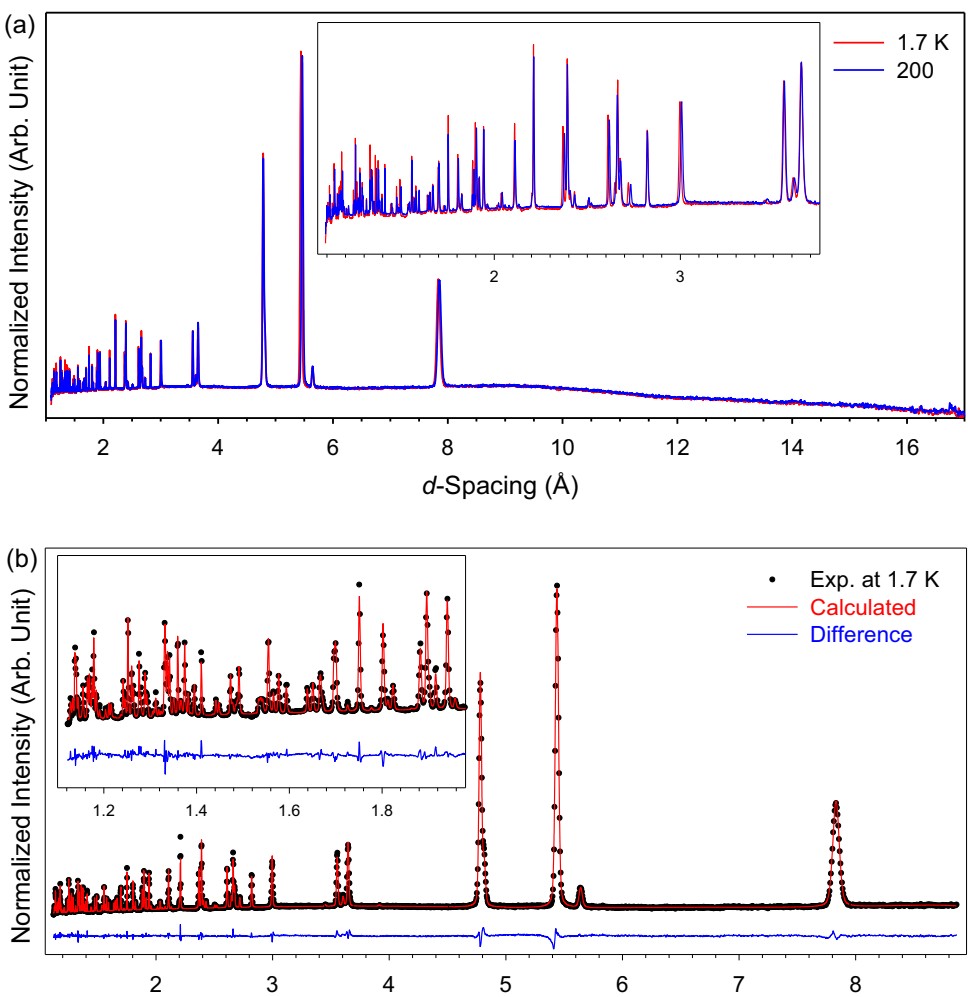

**Fig. 4 | PND patterns of NiBO and the Rietveld refinement of the PND data.**
**a** PND patterns of NiBO at 1.7 and 100 K. Patterns at 10 K, 20 K, and 100 K are
given in Supplementary Fig. 7. **b** Rietveld refinement of the PND data at 1.7 K;

$R_{\text{weighted profile}} = 2.49\%$; GOF (Goodness of Fit) = 5.08. Insets: low $d$-spacing regions.
Arb. unit = arbitrary unit. Source data are provided as a Source Data file.

with a Curie-Weiss temperature of $\theta_{\text{CW}} = -75.3(1)$ K, indicating anti-ferromagnetic interaction in NiBO. The Curie constant $C$ can be used to extract the effective magnetic moment $\mu_{\text{eff}}$ in units of the Bohr magneton, $\mu_{\text{B}}$[39].

$$\mu_{\text{eff}} = \sqrt{8C}\,\mu_{\text{B}} \tag{2}$$

giving $\mu_{\text{eff}} = 2.7(1)\,\mu_{\text{B}}$, which is comparable to the Ni$^{2+}$ spin-only value of $2.8\,\mu_{\text{B}}$[40]. The suppression of the DC magnetic susceptibility data at low temperatures is indicative of a finite magnetic excitation gap in such spin-1 chain systems[31,32]. The spin gap was extracted by fitting DC magnetic susceptibility data in the low-temperature range ($T < 50$ K) to an activated behavior:[3,31]

$$\chi(T) = \chi(0) + C \exp[-E_{\text{g}}/(k_{\text{B}}T)] \tag{3}$$

where $E_{\text{g}}$ is the spin gap and $\chi(0)$ is an offset.

Fig. 5b demonstrates an excellent fit of the experimental data to Eq. 3. We obtain a value of $E_{\text{g}} = 7.5(5)$ cm$^{-1}$, which is in overall agreement with the refined value of the spin gap extracted from QMC simulations, as discussed in the next section. In addition, the magnetization measurements show no increase in the magnetic susceptibility down to 2 K, suggesting that there is very little to no paramagnetic contamination of the Ni$^{2+}$ ions in NiBO.

To probe the stability of the gapped magnetic phase, we measured the DC magnetic susceptibility under various magnetic fields between 2 and 10 K (Fig. 5c). Note that at magnetic field ($H'$) ≤9 T, the magnetic susceptibility decreases with decreasing temperature. Above 9 T, an upturn occurs at low temperatures. In addition, at higher fields, the magnetic susceptibility starts to increase below 5 K, suggesting a possible magnetic phase transition. Supplementary Fig. 8 shows the field dependence of the DC magnetization between 2 K and 300 K, revealing nonlinear behavior below 5 K. The increase of magnetization signifies a possible field-induced phase transition. Background subtraction for the DC magnetic susceptibility data were performed through fittings of the data in the temperature range between 200 and 300 K using $\chi = C/(T - \theta_{\text{CW}}) + \chi_{\text{para}}$, where $\chi_{\text{para}}$ is the paramagnetic contribution in the background.

**Quantum Monte-Carlo simulations**
QMC simulations were performed in conjunction with the experimental DC magnetization measurements to identify the microscopic Haldane chain parameters for NiBO. We considered an isolated spin-1 chain model, described by the Hamiltonian:

$$H = J\sum_{i=1}^{N} S_i \cdot S_{i+1} + D\sum_{i=1}^{N}\left(S_i^z\right)^2 + E\sum_{i=1}^{N}\left[\left(S_i^x\right)^2 - \left(S_i^y\right)^2\right] \tag{4}$$

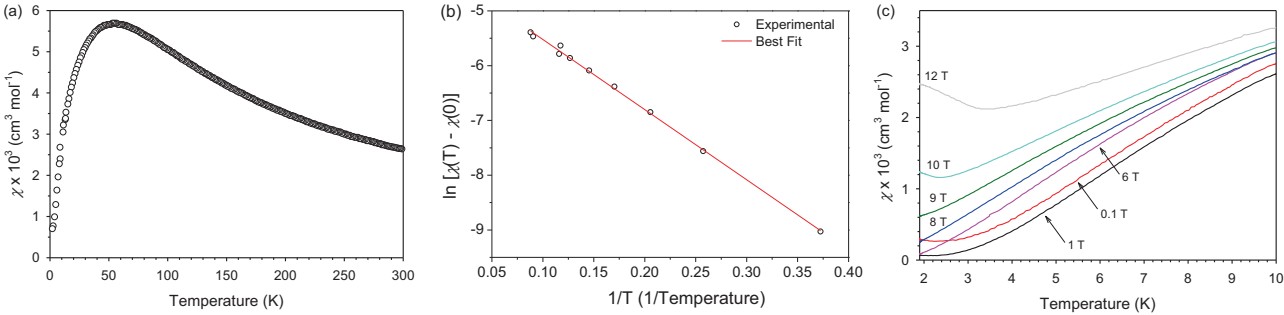

**Fig. 5 | DC magnetic susceptibility data of NiBO. a** Temperature dependence of the DC magnetic susceptibility ($\chi$) of NiBO at 0.1 T. **b** Plot of ln[$\chi$(T)--$\chi$(0)] vs. 1/T for data below 11.5 K. **c** Temperature dependence of the DC susceptibility of NiBO at magnetic fields of 0.1-12 T. Source data are provided as a Source Data file.

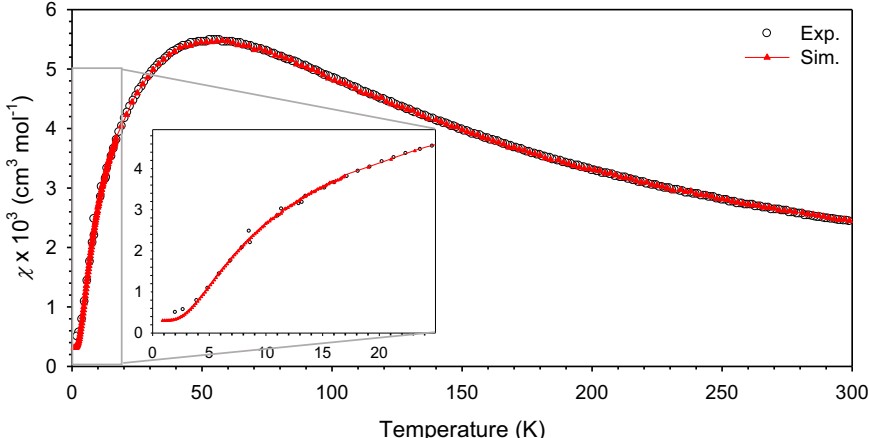

**Fig. 6 | Quantum Monte Carlo results for the magnetic susceptibility of NiBO as a function of temperature.** The inset shows the magnetic susceptibility data in the low-temperature range. The QMC data fits well with the experimental data down to the low-temperature regime. Source data are provided as a Source Data file.

where $J$ is the nearest-neighbor exchange coupling constant, $D$ is the axial single-ion anisotropy for Ni$^{2+}$, and $E$ is the transverse (or rhombic) anisotropy.

To perform the QMC simulations, we used the stochastic series expansion (SSE) method[41,42], and adjusted the above model parameters to obtain the best fit of the QMC magnetic susceptibility results to the experimental DC magnetic susceptibility data. The spin-1 chain was simulated with 512 sites and periodic boundary conditions to probe bulk properties. We note here the symmetry $E \rightarrow -E$. Therefore, we consider only $E > 0$ for the following. The Heisenberg coupling $J$, the $g$-factor, and the paramagnetic offset $\chi_{\mathrm{para}}$ of the susceptibility can be obtained by comparing the susceptibility data to the QMC simulations of the isotropic chain (with $D = E = 0$) as seen in Supplementary Fig. 11. We obtain $J = 29.44$ cm$^{-1}$, $g = 1.87$ and $\chi_{\mathrm{para}} \approx 10^{-5}$ cm$^3$/mol. The isotropic model agrees well with the experimental data at high temperatures but deviates at lower temperatures. Therefore, the isotropic model does not reproduce NiBO's low-energy spectrum. If the anisotropy $D < J$ is small and $|E| < D < J$, the full spectrum may be obtained using the following gaps[43,44] (where $\Delta_x < \Delta_y < \Delta_z$):

$$\Delta_x = 0.41J - 0.57D - \kappa_0 |E|$$

$$\Delta_y = 0.41J - 0.57D + \kappa_0 |E|$$

$$\Delta_z = 0.41J + 1.41D \qquad (5)$$

where $\kappa_0 = 2.05$ in our parameter regime.

To compare the QMC results to the experimental DC magnetic susceptibility data of the powder sample, we performed an orientational averaging, which yields the magnetic susceptibility:

$$\chi = \frac{2}{3}\chi_\perp + \frac{1}{3}\chi_\| + \chi_{\mathrm{para}} \qquad (6)$$

where $\chi_\perp$ ($\chi_\|$) denotes the QMC response perpendicular (parallel) to the direction of the anisotropy term and $\chi_{\mathrm{para}}$ is a paramagnetic contribution.

The results obtained from QMC simulations are compared to the experimental results in Fig. 6 where the $D$ and $E$ terms are included. We fixed $J$, $g_\perp = g_\|$, and $\chi_{\mathrm{para}}$ to the values obtained from the isotropic model, where $g_\|$ ($g_\perp$) is the $g$-factor for the magnetic fields applied parallel (perpendicular) to the direction of the anisotropy term [i.e., in a finite magnetic field with flux density $B$, the Zeeman terms $H_\| = g_\| \mu_B B \sum S_i^z$, and $H_\perp = g_\perp \mu_B B \sum S_i^x$ are added to the Hamiltonian ($H$), respectively]. The results show that with the inclusion of an additional $D$ and $E$ term, where $D = 4.99$ cm$^{-1}$ and $|E| = 1.01$ cm$^{-1}$, the simulation fits remarkably well to the experimental DC magnetic susceptibility data across the full temperature range from 1.841 to 300 K. The model parameters obtained from the simulations are listed in Table 1. Simulation results for different values of $|E|$ are compared to the experimental data in Supplementary Fig. 12. From $D$, the axial anisotropy parameter of one Ni$^{2+}$ ion, the effective anisotropy parameter for the chain (i.e., the collective triplet) $D_{\mathrm{eff}} = -1.98D = -9.88$ cm$^{-1}$ is obtained[29]. It should be pointed out that $D_{\mathrm{eff}} = -1.8D$ has also been used in the literature[15,45], which would make $D_{\mathrm{eff}}$ here smaller.

**Table 1 | Magnetic parameters from QMC simulations**

| Parameters | Energy (cm$^{-1}$) |
|---|---|
| $D$ | 4.99 |
| $D_{eff}$ | −9.88 |
| $|E|$ | 1.01 |
| $|E_{eff}|$ | 2.07 |
| $J$ | 29.44 |
| $\Delta_x$ | 7.16 |
| $\Delta_y$ | 11.3 |
| $\Delta_z$ | 19.1 |

The values of $D$ and $E$ are small enough for NiBO to reside within the Haldane phase rather than the large-$D$ phase[46,47]. Based on the QMC simulations[48], we obtain spin gaps of $\Delta_x = 7.16$ cm$^{-1}$, $\Delta_y = 11.3$ cm$^{-1}$, and $\Delta_z = 19.1$ cm$^{-1}$. In addition, QMC results for the magnetization in finite fields are shown in Supplementary Figs. 18 and 19. Similar to the experimental magnetic susceptibility data obtained in finite magnetic fields shown above, the finite-field QMC data demonstrate the stability of the Haldane phase for magnetic field strength up to the size of the zero-field spin gap. Additional results for the magnetic field dependence of the excitations gaps for the powder sample are also provided in the SI.

**Specific heat**

Specific-heat ($C'$) measurements on NiBO were conducted to complement the magnetic susceptibility data. Supplementary Fig. 20 shows the specific-heat data between 1.8 K and 200 K at zero field and in a magnetic field ($H'$) of 12 T for a powder sample of NiBO. For the magnetic field $H'$ beyond 9 T, a $\lambda$ shape peak shows up below 5 K and shifts toward a higher temperature as the magnetic field increases, as shown in Fig. 7a. The peaks indicate a field-induced phase transition occurring in the above field and temperature range. A summary of the critical field ($H'_N$) and critical temperature ($T_N$) of the transition line is shown in Supplementary Fig. 21. The transition points observed in both the specific-heat data and the magnetization data are consistent with each other, indicating that there is a single magnetic phase transition in NiBO at a magnetic field ($H'$) beyond 9 T and below 5 K. The change in the critical field and critical temperature of the transition is almost linear, which is similar to previous observations in NENP[49]. In gapped quantum spin systems, field-induced Bose-Einstein condensation beyond the critical field strength was predicted[50–52] and experimentally observed previously in Ni$^{2+}$-based spin-1 compounds[53,54]. Namely, above the critical field $H'_N$, magnon excitations condensate at low temperatures. Quantitatively, one would expect the field-induced transition temperature $T_N$ proportional to $(H'-H'_N)^\varphi$, where the crossover exponent $\varphi$ depends on the dimensionality and the magnon dispersion relation[55,56]. In the inset of Fig. 7b, we replot our data as $\ln T_N$ vs $\ln (H'-H'_N)$. By fitting the data to $T_N$ vs $(H'-H'_N)^\varphi$, $\varphi = -0.54(6)$ and $H'_N = 7.7(4)$ T are obtained. The extracted $\varphi$ is too small for either effective dimensionality ($\varphi = 2/3$ for 3D and 1 for 2D). Measurements at lower temperatures are necessary for a more accurate determination of the critical exponent.

**Inelastic neutron scattering (INS) at 0 T**

From the energy diagram of a Haldane spin-1 chain system and the Zeeman effect on the magnetic states in Fig. 1, two transitions, A and B in Fig. 8, are expected at 0 T.

INS was conducted at VISION without a magnet to study the spin gaps in NiBO-$d_8$. The deuterated sample was used, as the scattering cross section of D atoms is much smaller than that of H atoms[57], significantly enhancing signal/noise ratios of magnetic transitions in INS spectra of NiBO-$d_8$ in comparison with spectra of H-containing NiBO. The measurements were performed at 5, 7.5, 10, 15, and 20 K to

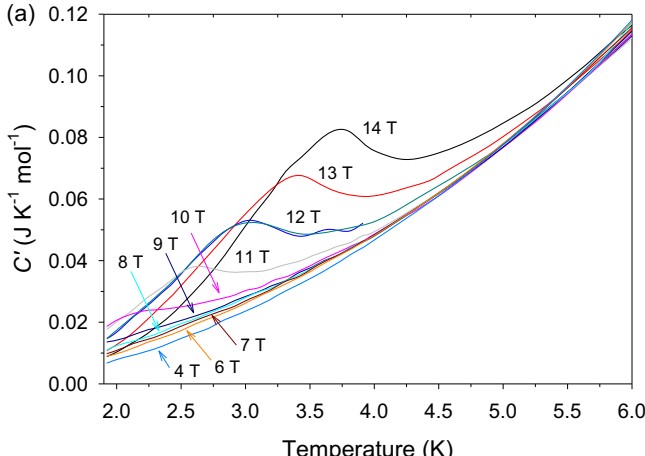

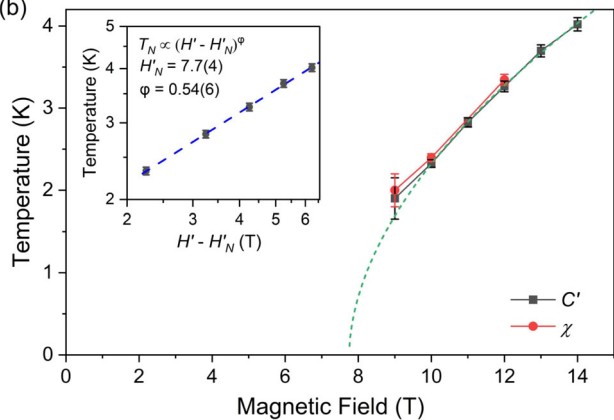

**Fig. 7 | Specific heat data and the temperature-magnetic field phase diagram of NiBO showing a phase transition. a** Magnetic field-dependent specific heats of NiBO from 1.92 to 6.00 K at 0-14 T. **b** Temperature-magnetic field phase diagram of NiBO from specific heat ($C'$) and magnetization ($\chi$) including their respective error bars in the temperatures. The green dash line presents the fitting $T_N$ vs $(H'-H'_N)^\varphi$. Inset: Plot of $\ln T_N$ vs $\ln (H'-H'_N)$ including error bars in $\ln T_N$. $T_N$ is the critical temperature. $H'_N$ is the critical field. $H'$ is the magnetic field. $\varphi$ is the crossover exponent. Source data are provided as a Source Data file.

determine the spin gaps between the singlet ground state and the triplet excited state. Two different spin gaps are observed at 5 K in Fig. 9. The intensities of the two peaks at ~10 and 19 cm$^{-1}$ decreased with increasing temperature. The temperature-dependent of the peaks suggest a magnetic origin and not phonons. The peak at 10.0(1) cm$^{-1}$ is assigned to originate from the transition A, $|0,0\rangle \rightarrow |1,\pm1\rangle$, based on the spin gap $E_g = 7.5(5)$ cm$^{-1}$ from susceptibility data which agrees with that from the QMC calculations. The peak at 19.1(1) cm$^{-1}$ originates from the transition B, $|0,0\rangle \rightarrow |1,0\rangle$. The value of the experimental $D_{eff}$(INS) and the single-ion anisotropy $D$(INS) can be determined using the magnetic coupling constant ($J$) from QMC (29.44 cm$^{-1}$) and peak B [$\Delta_z = 19.1(1)$ cm$^{-1}$] from INS in Eqs. 7 and 8:[15,22]

$$D(INS) = \frac{\Delta_z - 0.41J}{1.41} = 5.0(1) \text{ cm}^{-1} \quad (7)$$

$$D_{eff}(INS) = -1.98D(INS) = -9.9(1) \text{ cm}^{-1} \quad (8)$$

We are unable to observe the splitting of peak **A** ($\Delta_{xy}$) due to the limited resolution of the instrument. With higher resolution, peak A should split into $\Delta_x$ and $\Delta_y$ indicating an existence of the transverse or rhombic anisotropy ($E$) in NiBO. The results obtained from INS agree well with the QMC and magnetic susceptibility results.

## High-field electron spin resonance

HFESR measurements were performed at 4.5 K on a powder sample of NiBO (0.116 g) to probe the electronic spin transition within the triplet excited state. The results show only a single low-intensity and broad resonance in Fig. 9b at ~6.8 T and 511 GHz. The resonance moves to lower magnetic field upon decreasing the frequency, as shown in Supplementary Fig. 10. We interpret the observed resonance as originating from the transition $|1, -1\rangle \rightarrow |1,0\rangle$. A simple linear extrapolation to 0 T gives a lower bound estimated $D_{eff}(\text{HFESR}) = 7.8(2)$ cm$^{-1}$. However, this value underestimates the actual zero-field gap, as discussed in the SI. Instead, the results obtained based on the QMC simulations compare well with the experimental data and support the observed transition $|1, -1\rangle \rightarrow |1,0\rangle$ in the powder averaged spectrum of the HFESR, as shown in Supplementary Fig. 17. Furthermore, the intensity of the resonance is weak, probably resulting from the low thermal population of the triplet state at 4.5 K. The even lower population of the $|1, +1\rangle$ state may furthermore explain the absence of the resonance transition $|1, +1\rangle \rightarrow |1,0\rangle$ in the spectrum. The result is consistent with observations in the ESR spectra of PbNi$_2$V$_2$O$_8$[58]. No resonances are observed for the formally forbidden transitions from the singlet to the triplet state, based on the selection rules. The end-chain resonance is not discernible in the spectrum due to NiBO having little to no impurities in the MOF structure as well as negligible inter-chain coupling. According to Čižmár et al., observation of the end-chain resonance can be attributed to the spin-1/2 fractional end-chain effect or the presence of the inter-chain coupling[22]. Impurities can cause fragmentation of the spin-1 chains leading to the presence of more end-chain spin-1/2. As determined in the magnetization measurements of NiBO, the magnetic susceptibility does not increase at low temperatures, which is the characteristic of a relatively pure system.

## Phonon spectrum from INS and comparison with the DFT-calculated INS spectrum

The phonons described in these studies are those of both inter- and intra-molecular vibrations. INS is based on the neutron kinetic energy transfer, which means the technique does not follow any selection rules, unlike optical spectroscopies such as IR or Raman. Therefore, INS can detect all the phonon excitations in the solid molecule[59]. Supplementary Fig. 22 shows the experimental INS spectra of NiBO and NiBO-$d_8$ at 5 K and the calculated phonon spectra from DFT calculations. For NiBO, most of the major phonon features are located below 1700 cm$^{-1}$, while one distinct major feature is at 3100 cm$^{-1}$. The calculated phonon spectra fit well to the experimental phonon spectra, especially in the higher energy region. Further discussion of the phonon features, phonon symmetries, and spin densities are provided in SI.

Many previous studies have focused on studying 1D Haldane chains in 3D structures, but rarely in a 2D material. Here, we have provided a thorough characterization of the magnetism in the 2D-MOF compound NiBO, which consists of stacked 2D planes of parallel spin-1 chains, formed by Ni$^{2+}$ ions. From our combined analyses, using a wide range of experimental probes, including variable-temperature powder neutron diffraction (VT-PND), DC magnetic susceptibility, specific-heat, and HFESR measurements, we identify NiBO as a clean realization of the Haldane spin-1 chain with weak interchain coupling, due to the stacked planar crystal structure and a twisting in the bipyridine ligands that stabilize the planar arrangement. This sets aside NiBO from various previously explored Haldane spin-1 chain compounds, which are mostly imbedded in a 3D crystal structure. VT-PND confirms that NiBO does not exhibit long-range antiferromagnetic order, supporting that NiBO is a Haldane spin-1 chain material with short-range spin

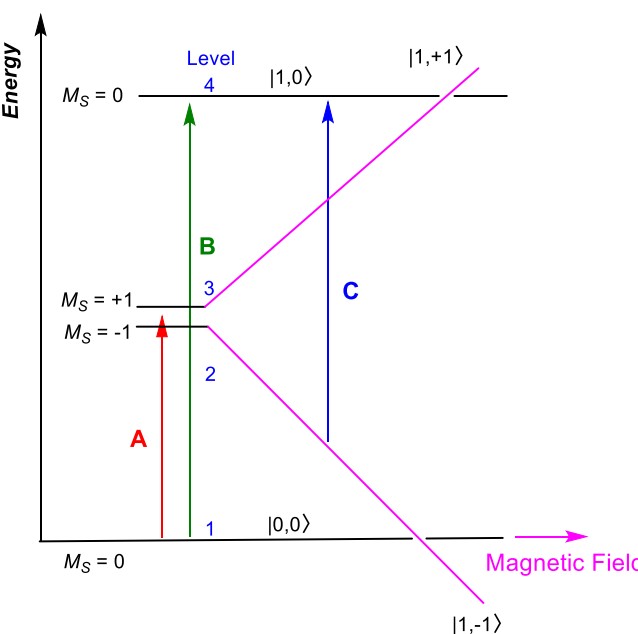

**Fig. 8 | Energy diagram of NiBO showing three observed magnetic transitions.** Expected transitions A, B at 0 T and C inside magnetic fields for a single crystal of NiBO with the magnetic field aligned in the $z$-direction. The $M_S = +1$, −1, and 0 states are defined in Fig. 1.

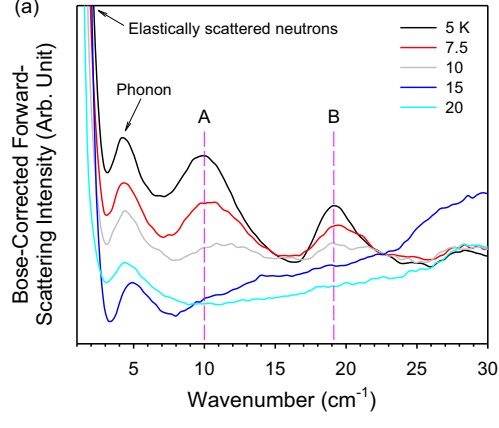

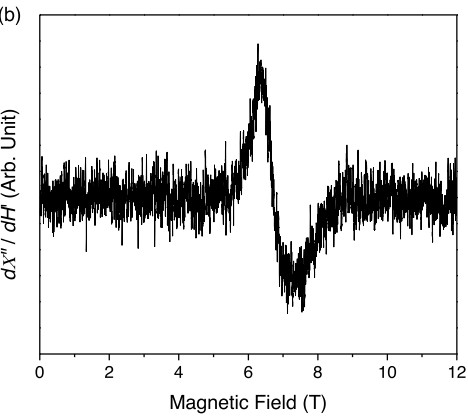

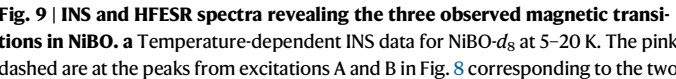

**Fig. 9 | INS and HFESR spectra revealing the three observed magnetic transitions in NiBO. a** Temperature-dependent INS data for NiBO-$d_8$ at 5–20 K. The pink dashed are at the peaks from excitations A and B in Fig. 8 corresponding to the two spin gaps between levels 1 and 2/3 as well as between levels 1 and 4. **b** HFESR spectrum of NiBO at 4.5 K and 511 GHz. $dX''/dH'$ is the first derivative of the absorption $X''$ vs magnetic field $H'$. Source data are provided as a Source Data file.

correlations and a disordered ground state. Furthermore, specific-heat and DC magnetic susceptibility experiments indicate a phase transition induced by the magnetic fields beyond the gap-closing field of 9 T and at temperature below 5 K. We have conducted key in-depth analysis of a 2D-MOF with decoupled spin-1 chains exhibiting Haldane physics, comparing remarkably well to a microscopic spin-1 chain model using unbiased QMC simulations. Parameters from QMC simulation of the microscopic spin model for NiBO are $J = 29.44$ cm$^{-1}$, $D = 4.99$ cm$^{-1}$, $D_{\text{eff}} = -9.88$ cm$^{-1}$, $|E| = 1.01$ cm$^{-1}$, $E_{\text{eff}} = 2.07$ cm$^{-1}$, $\triangle_x = 7.16$ cm$^{-1}$, $\triangle_y = 11.3$ cm$^{-1}$, and $\triangle_z = 19.1$ cm$^{-1}$. INS measurements provide the energy gaps between the singlet ground state and two levels of the excited triplet state in Fig. 9a to be 10.0(1) cm$^{-1}$ and 19.1(1) cm$^{-1}$, respectively. In addition, INS studies give $D_{\text{eff}}(\text{INS}) = -9.9(1)$ cm$^{-1}$; $D(\text{INS}) = 5.0(1)$ cm$^{-1}$. HFESR probe the effective anisotropy of NiBO with a lower bound for $D_{\text{eff}}(\text{HFESR}) = 7.8(2)$ cm$^{-1}$. MOFs are unique among Haldane spin-1 chain materials by providing the possibility for exfoliation into 2D nanosheets with unique magnetic properties. Our works open interesting new directions for future research on low-dimensional quantum magnetism in metal-organic frameworks.

## Methods

Syntheses of NiBO and NiBO-$d_8$ are given in Supplementary Information.

### Single-crystal X-ray diffraction

Single-crystal X-ray diffractionn (SCXRD) data were collected using a Bruker D8 Venture at 100 K with Mo Kα radiation. Data were collected and integrated using APEX 3 programs, reduced using Bruker SAINT program, and corrected for absorption using the SADABS multi-scan program. The structure was solved using SHELXT and refined with SHELXL-2015. There was no distinction made for the D isotope in NiBO-$d_8$[60].

### Magnetic susceptibility (DC) and specific heat

DC magnetic susceptibility measurements at 0.1 T were conducted at the Center for Nanophase Materials Sciences (CNMS) at Oak Ridge National Laboratory (ORNL) using the superconducting quantum interference device (SQUID) magnetometer by Quantum Design. A powder sample of NiBO with a total mass of 119.2 mg was used. Variable temperature and variable field DC magnetic susceptibility were measured using the Physical Property Measurement System (PPMS) Dynacool by Quantum Design with the Vibrating Sample Magnetometer (VSM) option. The powder with a total mass of 11.7 mg was used for the temperature-dependent of magnetization under a magnetic field of 0.1 T from 2 K to 300 K. The magnetization was performed between 2 K and 300 K up to 14 T. Temperature-dependent of specific heat was measured by relaxation technique using the PPMS. A pressed pellet with a total mass of 6.5 mg was used for specific-heat measurement between 2 K to 200 K from 0 T to 14 T.

### High-field electron spin resonance

HFESR experiments were performed at the National High Magnetic Field Laboratory (NHMFL) using a homemade ESR spectrometer with a 17 T superconducting magnet[61]. A low-frequency source (13.5–18.5 GHz) was used in conjunction with an array of multipliers and amplifiers (Virginia Diodes, Charlottesville, VA, USA) to generate higher frequency harmonics.

### Powder neutron diffraction

Neutron diffraction data were collected at the POWGEN diffractometer at the Spallation Neutron Source (SNS) at Oak Ridge National Laboratory (ORNL) between 1.8 K and 200 K using an Orange Cryostat. A powder sample was loaded into a 6 mm-diameter, cylindrical vanadium sample can, and the data were collected for approximately 2 h in the high-resolution mode, using a center wavelength of 2.665 Å

covering the $d$ spacing from 1 Å to 15.0 Å. A background correction for the measured data was performed using an empty vanadium sample can measurement. The peak profile was described using a convolution of a Gaussian peak shape and a GSAS back-to-back exponential peak shape accounting for the asymmetry. The peak profile was obtained by refining data from Si (SRM 640d, NIST). All refinement was carried out using the TOPAS 6 software of the academic version[62].

### Inelastic neutron scattering

VT INS data were collected by placing a powder sample of NiBO (-0.5 g) or NiBO-$d_8$ (-1.0 g) the into a vanadium can specifically made for the neutron instrument. The sample vanadium can was placed into the neutron beamline at the Vibrational Spectrometer (VISION) at SNS, ORNL. There are two detector banks for the forward (low $|\mathbf{Q}|$) and the back (high $|\mathbf{Q}|$) scattering of neutrons. The effect of phonon population was accounted for through normalization of the INS intensity at energy transfer ω with $\coth\left(\frac{\hbar\omega}{2k_B T}\right)$[63].

### DFT calculations of phonon and spin density

Spin-polarized density-functional theory (DFT) calculations of NiBO and NiBO-$d_8$ were performed at SNS, ORNL using the Vienna Ab initio Simulation Package (VASP)[64]. The calculation used Projector Augmented Wave (PAW) method[65,66] to describe the effects of core electrons, and Perdew-Burke-Ernzerhof (PBE)[67] implementation of the Generalized Gradient Approximation (GGA) for the exchange-correlation functional. The energy cutoff was 800 eV for the plane-wave basis of the valence electrons. The lattice parameters and atomic coordinates from the CIF file, generated by the SCXRD measurement of NiBO-$d_8$ at 100 K, were used as the initial structure. In the calculations for NiBO, the D atoms in NiBO-$d_8$ were replaced by H atoms. The electronic structure was calculated on a $3 \times 3 \times 7$ Γ-centered mesh. The total energy tolerance for electronic energy minimization was $10^{-8}$ eV, and for structure optimization, it was $10^{-7}$ eV. The maximum interatomic force after relaxation was below 0.001 eV/Å. The optB86b-vdW functional[68] for dispersion corrections was applied, and a Hubbard U term of 6.2 eV[69] was applied to account for the localized 3$d$ orbitals of Ni. A $1 \times 1 \times 3$ supercell was created, for which the electronic structure was calculated on a $3 \times 3 \times 2$ Γ-centered mesh. The interatomic force constants were calculated on the supercell by VASP, and the vibrational eigenfrequencies and modes were then calculated using Phonopy[70]. The OCLIMAX software[71] was used to convert the DFT-calculated phonon results to the simulated INS spectra.

## Data availability

Source data are provided with this paper. The data generated in this study have been deposited in the Figshare.

## Code availability

VASP (Vienna ab initio simulation package) for the periodic DFT phonon calculations is available at https://www.vasp.at/. QMC codes are available on request.

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

## Acknowledgements

The US National Science Foundation (CHE-2055499 to Z.-L.X.) and a Shull Wollan Center Graduate Research Fellowship (to P.T.) are acknowledged for partial support of the research. Part of this work was performed at the National High Magnetic Field Laboratory which is supported by NSF Cooperative Agreement No. DMR-1644779 and the State of Florida. Magnetic property study was conducted in part at the Center for Nanophase Materials Sciences, which was sponsored at Oak Ridge National Laboratory by the Scientific User Facilities Division, Office of Basic Energy Sciences, U.S. Department of Energy. Neutron scattering experiments were conducted at ORNL's Spallation Neutron Source, which is supported by the Scientific User Facilities Division, Office of Basic Energy Sciences (BES), U.S. Department of Energy (DOE), under Contract No. DE-AC0500OR22725 with UT Battelle, LLC. The computing resources were made available through the VirtuES and the ICEMAN projects, funded by Laboratory Directed Research and Development program and Compute and Data Environment for Science (CADES) at ORNL. We acknowledge support by the Deutsche Forschungsgemeinschaft (DFG) through Grant No. WE/3649/4-2 of the FOR 1807 and RTG 1995 (to S.W.), and thank the IT Center at RWTH Aachen University and JSC Jülich for access to computing time through JARA CSD. The authors thank Dr. Phattananawee Nalaoh for help with SCXRD and Dr. Andrzej Ozarowski for help with HFESR interpretation. Department of Chemistry and Open Publishing Support Fund at the University of Tennessee-Knoxville are acknowledged for open access to this research.

## Author contributions

P.T. performed syntheses of NiBO and NiBO-$d_8$, grew single crystals of NiBO-$d_8$, prepared all samples for the experiments indicated below, and coordinated the research. M.J.J. performed the SCXRD experiment and solved the crystal structure. M.J.J. and P.T. analyzed and interpreted the data. Z.G. performed the magnetic susceptibility measurement at a low magnetic field (0.1 T). P.T. analyzed and interpreted the data. J.X. performed the measurements of magnetic susceptibilities at high magnetic fields (>0.1 T) and specific-heat capacity. J.X., P.T., and R.J. analyzed and interpreted the data. R.J. supervised the studies here. N.C. and S.W. performed the quantum Monte-Carlo simulations for the magnetic susceptibility and the specific-heat capacity data. N.C., S.W., and P.T. analyzed and interpreted the results. J.K. performed the HFESR measurement. P.T. and J.K. analyzed the data and interpreted the results. C.L. performed PND experiments at POWGEN. C.L. and P.T. analyzed and interpreted the data. L.L.D. performed INS experiments at VISION. P.T. and L.L.D. analyzed and interpreted the data. Y.C. performed the VASP calculations. P.T. analyzed and interpreted the data. Z.-L.X. initiated the project, supervised the research, including data analyses and interpretations, and coordinated the studies.

## Competing interests

The authors declare no competing interests.
