## [Peer Review File · Nature Communications]

Haldane topological spin-1 chains in a planar metal-organic frameworkREVIEWER COMMENTS

Reviewer #1 (Remarks to the Author):

Notes

In this paper Tin et al investigate the magnetic properties of nickel(II) oxalate bipyridine and show it is a Haldane $S=1$ magnet. The paper is very thorough and well executed, with a full set of characterisation techniques and they convincingly demonstrate the presence of the topological Haldane phase in this Ni(II) oxalate coordination polymer. It should be published, but the key questions I was left with after reading this paper are:

- How does this material differ from the large number of known Haldane $S=1$ material?
- and/or what is the new physics uncovered in this material?

To expand:

- The compound has been previously reported, although it appears to perhaps be in a different polymorph and its Haldane properties missed.
- Other Ni(ox)₂ 2D polymers show Haldane physics, and though the authors here show high quality characterisation absent from other papers there is little demonstration of the
- $S=1$ Haldane physics is relatively well understood and this compound provides strong evidence of the validity of the canonical description. I could not clearly see where this

One point of novelty emphasised by the authors is that this material is a 2D, not 3D. The characterisation of this material as 2D relies on the crystal structure: which would also suggest that other Haldane materials tend to be 1D not 3D, and would also apply to many of the other Ni oxalates. It is unclear what consequences the authors anticipate this 2D-ness entailing and the authors do not do any measurements exploiting this facet.

In conclusion, this paper is high quality science carefully done, but explicit statement of where it advances our understanding would be useful.

Some additional less critical points

- Single crystal data is very good, but should be refined using inversion twins rather than with a large flack parameters.
- The introduction and discussion should emphasise that the material is known. Reference 38 should be more prominent, especially considering the authors even used its synthesis method.
- PXRD appears to show minor impurities (e.g. at 12, 33°). The authors should discuss if not correct and probably do a Rietveld refinement or at least a quick Pawley
- The results of the Rietveld refinement at 1.8K are not shown, and it's hard to see what's going on in the high Q region. As the authors have used the beam time, they should show more than a figure!
- The authors description of the previous structure is somewhat confused in the SI: the difference in the unit cell angle is mostly due to choice of setting: the equivalent angle in the Immm structure will be much closer to 116°, though the authors should calculate it properly.

Reviewer #2 (Remarks to the Author):

The authors of this manuscript present a comprehensive study of the magnetic properties of a novel spin-1 Haldane compound. This material is remarkable because, as opposed to previously studied spin-1 compounds, it is effectively two-dimensional. The experimental measurements are complemented by theoretical calculations that consistently confirm that this material can be effectively modeled as an anisotropic spin-one Heisenberg quantum spin chain, characterized by a spin gap that can be overcome by applying an external magnetic field. I find this study interesting and appealing, but I would like the authors to clarify one set of issues before agreeing to publish it.

Multiple indications are reported for a magnetic field induced transition, consistent with the Haldane picture. However, no attempt is made to further characterize the high field regime, e.g., by looking for possible field induced off-diagonal long-range order etc. Moreover, one could attempt to extract, or at least comment on, the critical exponents associated with this phase transition, which could possibly be a type of magnetic Bose-Einstein condensation, as has previously been discussed in the context of NENP.

Reviewer #3 (Remarks to the Author):

The work by Tin et al. provides an interesting comprehensive study of what appears to be a previously known 2D coordination polymer consisting of chains of Ni oxalates bridged by bipyridyl ligands. The substantial claim that the author make is that this compound constitutes a rare 2D MOF example of Spin-1 Haldane chain compound which has interesting applications since 2D MOFs can be exfoliated to provide single-layer 2D materials for nanomagnetism.

The general claim is correct, there are not many examples of such systems, and the compound is indeed interesting.

However, two points are worth mentioning, i) the compound itself is already known as presented by the authors (ref. 38), therefore the compound itself is not a novelty, ii) as far as properties go, the properties situate NiBO indeed as part of a rare ensemble of 2D coordination polymer systems exhibiting Spin-1 Haldane chains, but in this particular case there is essentially no element of surprise since there are at least four precedents (refs. 31, 32, 33) of compounds showing essentially the same properties, with ligands (bpe, bpa and piperidine) that are extremely similar (especially true for bpe and bpa). In other words, the results described by the authors, although quite thorough and well-executed (in the most parts) are pretty much in line with what one would expect based on the known literature precedents and the minor modifications in the ligand structure. As such, my opinion is that this beautiful study is not novel enough to warrant publication in nature communications, or if it is, the authors failed to convey that novelty to a reader such as myself.

Beyond this fact, here are a few points that the authors could address to improve their manuscript:

- There are confusions between NiBO and NiBO-d8. It is not always very clear what has been done on NiBO and on NiBO-d8.
- The Structural information of NiBO-d8 at 100K is labelled as NiBO in table S1, or is it really NiBO? It seems to be the result of SCXRD on NiBO-d8. This point could be clarified.
- If this is indeed NiBO-d8, comparing structural information on the deuterated analog at 100K to structural information on the hydrogenated analogue at 295K is not enough. The difference in ring conformation may be due to the deuteration and not to temperature. This can easily be tested by measuring the d8 compound at 295K and see if it is in the orthorhombic or monoclinic phase.
- On p. 11 it appears that there is a problem in the description of Fig.6. I don't see as the author claims that "X decreases then increases" as a function of increasing magnetic field. It seems to be true if one looks at the background corrected data on Fig. S9, but not in Fig. 6.
- Fig S21 is labelled as Fig. 21.
- P.18: "the peak at 10.0 cm⁻¹ is determined to originate from the transition..." how were the assignment done? a quick explanation would probably suffice.
- I may have missed a point but why is the INS data from fig. S22 different from those of Fig.10 at 5K ?

RESPONSE TO REVIEWERS' COMMENTS

We thank the reviewers for the comments. Revisions have been made to the manuscript to address reviewers' comments. The following point-by-point response to reviewers' comments details the changes in the revised manuscript. The response is in red color.

Reviewer #1

- How does this material differ from the large number of known Haldane $S = 1$ material?
- and/or what is the new physics uncovered in this material?

To expand:

- Little demonstration of how this material is different or better than previous exemplar materials.
- $S = 1$ Haldane physics is relatively well understood and this compound provides strong evidence of the validity of the canonical description. I could not clearly see where this material brings new insights or poses new questions.

The reviewer is correct that Haldane $S = 1$ materials have been reported. The comprehensive studies of **NiBO** in the current work offer the following new science:

- (1) Haldane chains in **NiBO** are neutral. In contrast, most reported Haldane $S = 1$ chains are cationic in, e.g., NENP, NDMAZ, and $[\text{Ni}(\text{HF})_2(\text{pyz})_2]\text{SbF}_6$. This is because in NiBO, ox^{2-} dianion balances the +2 charge on Ni^{2+} , while in many reported Haldane chains, -1 ligands (such as NO_2^- , N_3^- , or HF_2^-) bridge Ni^{2+} ions, requiring additional anions ClO_4^- , BF_4^- or SbF_6^- to balance the + charges on the chains. Thus, the use of -2 oxalate ligand making neutral Haldane $S = 1$ chains in **NiBO** is novel. Although four other Ni^{2+} chains with ox^{2-} (cited in p. 5, middle) have been reported to show the presence of a spin gap, the evidence is from DC magnetic susceptibility measurements. The current work on **NiBO** firmly demonstrates that -2 oxalate ligand, linking Ni^{2+} ions, does give neutral Haldane chains.
- (2) Unlike other Haldane $S = 1$ chains with *no* chemical bonds among the chains, the Haldane chains in **NiBO** are chemically linked by 4,4'-bipyridine ligands. The current work convincingly shows that the chemical links among the chains still let them keep

Haldane properties. The links, however, lead to the 2D structure in **NiBO** with relatively weak Van der Waals force between the layers, making it feasible to exfoliate the 2D-MOFs into nanosheets with interesting low-dimensional properties. In contrast, reported Haldane chains typically are cationic and cannot be exfoliated.

These two features are summarized in p. 5.

One point of novelty emphasised by the authors is that this material is a 2D, not 3D. The characterisation of this material as 2D relies on the crystal structure: which would also suggest that other Haldane materials tend to be 1D not 3D, and would also apply to many of the other Ni oxalates. It is unclear what consequences the authors anticipate this 2D-ness entailing and the authors do not do any measurements exploiting this facet.

The reviewer is correct that other Haldane materials tend to be 1D cationic, requiring anions to balance the charge. We have revised the statements in the manuscript, clarifying the point on p. 5. Also, we anticipate future studies to exfoliate the 2D MOF and then probe the 2D layer by scanning electron microscopy (SEM) and other surface techniques. For an exfoliated layer, studies of each Haldane chain may be performed. Quantum properties of each Haldane chain may lead to unique applications. Such studies are, however, outside the scope of the current work.

In conclusion, this paper is high quality science carefully done, but explicit statement of where it advances our understanding would be useful.

In addition to the new statements on p. 5 indicated above, we think detailed studies of the unusual -2 oxalate ligand in **NiBO**, leading to the Haldane-type antiferromagnetic coupling between two unpaired electrons of neighboring Ni^{2+} ions (AKLT model in Fig. 2), should be explored by theoretical studies to understand further the nature of the bonding, in addition to exploring the surface properties.

Some additional less critical points

- Single crystal data is very good, but should be refined using inversion twins rather than with a large flack parameters.

Twin law has been used as part of the refinement to obtain the single-crystal structure, as shown in the following part of the submitted cif:

```
_computing_data_reduction          ?
_computing_molecular_graphics      'Olex2 1.5 (Dolomanov et al., 2009) '
_computing_publication_material    'Olex2 1.5 (Dolomanov et al., 2009) '
_computing_structure_refinement    'SHELXL 2018/3 (Sheldrick, 2015) '
_computing_structure_solution      'olex2.solve 1.5 (Bourhis et al., 2015) '
_refine_diff_density_max           0.465
_refine_diff_density_min           -0.385
_refine_diff_density_rms           0.050
_refine_ls_abs_structure_details
;
  Refined as an inversion twin.
;
_refine_ls_abs_structure_Flack     0.491(16)
_refine_ls_extinction_coef         .
```

- The introduction and discussion should emphasise that the material is known. Reference 38 should be more prominent, especially considering the authors even used its synthesis method.

We have revised the manuscript:

- (1) In the first sentence in Results (p. 7), we added the following sentence “**NiBO** and its crystal structure at 295 K was reported earlier.³⁷”
- (2) In the first paragraph on **NiBO**, which is in Introduction, Ref. 37 is added was added to the sentence: “Here, we report that $[\text{Ni}(\mu\text{-}4,4'\text{-bpy})(\mu\text{-ox})]_n$ (**NiBO**)³⁷ is a Haldane spin-1 chain material...”.

- PXRD appears to show minor impurities (e.g., at 12, 33°). The authors should discuss if not correct and probably do a Rietveld refinement or at least a quick Pawley.

The reviewer is correct that a minor impurity was observed, for instance, at $d = 7.01 \text{ \AA}$ and $d = 4.32 \text{ \AA}$, in the powder X-ray diffraction (PXRD) data of **NiBO** (Fig. S1). (We have converted the PXRD plots in Figs. S1-S2 from 2θ (°) to d -spacing (Å) so they can be compared directly with those from PND in Fig. 4.) In Fig. S1, the impurity peaks have been now labeled. In the powder neutron diffraction (PND) experiment, a very diffused hump was observed at the same d -spacings. The intensities from the impurity are very low and are therefore not included in any of the structural model.

Fit of the powder X-ray diffraction data by the Pawley method has been performed and is now added to Fig. S1 (and Fig. S2 for **NiBO-d₈**) in the supplementary information (SI).

In addition, we have added more powder neutron diffraction (PND) data for **NiBO** at 10 K, 20 K, and 100 K in Fig. S7 for comparison with those at 1.7 K and 200 K. An enlarged, expanded *d*-spacing region is given in Fig. S7.

- The results of the Rietveld refinement at 1.8 K are not shown, and it's hard to see what's going on in the high *Q* region. As the authors have used the beam time, they should show more than a figure!

The results of the Rietveld refinement have been added to Fig. 4-Bottom caption. In addition, the high *Q* region has been added as insets in Fig. 4.

- The authors description of the previous structure is somewhat confused in the SI: the difference in the unit cell angle is mostly due to choice of setting: the equivalent angle in the *Immm* structure will be much closer to 116°, though the authors should calculate it properly.

We agree with the reviewer and have deleted the sentence: “The β angle of the unit cell in the 100 K structure deviated from 90° to ~116° placing the crystal system of the 100 K structure into the monoclinic (*C2*) system while the 295 K structure is in the orthorhombic (*Immm*) system.” The discussion on the two structures is now as follows (p. S-6):

“The reported crystal structures at 295 K and **NiBO-d₈** at 100(2) K are mostly similar with some major differences, such as solving the structure of **NiBO-d₈** in the monoclinic *C2* (No. 5) space group instead of the orthorhombic *Immm* (No. 71).”

Reviewer #2

I find this study interesting and appealing, but I would like the authors to clarify one set of issues before agreeing to publish it.

Multiple indications are reported for a magnetic field induced transition, consistent with the Haldane picture. However, no attempt is made to further characterize the high field regime, e.g., by looking for possible field induced off-diagonal long-range order etc. Moreover, one could attempt to extract, or at least comment on, the critical exponents associated with this phase transition, which could possibly be a type of magnetic Bose-Einstein condensation, as has previously been discussed in the context of NENP.

We highly appreciate Reviewer 2's point on the magnetic field effect in both the magnetization and specific heat, which clearly show the field-induced phase transition. In the original manuscript, we did not expand our discussion due to the narrow regions of both temperature and magnetic field of the phase diagram in the new Fig. 8-Bottom. In an $S = 1$ system, Bose-Einstein condensation was predicted and experimentally observed. With the applied magnetic field H' , the excitation energy of triplons ($S = 1$) is lowered and eventually crosses zero at critical field $H' = H'_N$. Above H'_N , magnon excitations condensate at low temperatures. Quantitatively, one would expect the field-induced transition T_N proportional to $(H' - H'_N)^\varphi$, where the exponent φ depends on the dimensionality and dynamical critical exponent.⁵³ In the inset of Fig. 8-Bottom, we plot the data as $\ln T_N$ versus $\ln (H' - H'_N)$. By fitting the data to T_N proportional to $(H' - H'_N)^\varphi$, we obtain $\varphi = \sim 0.54(6)$ and $H'_N = 7.7(4)$ T.

In the revised manuscript (p. 17-middle), we have added the following:

“In gapped quantum spin systems, field-induced Bose-Einstein condensation beyond the critical field strength was predicted⁵¹⁻⁵³ and experimentally observed previously in Ni-based spin-1 compounds.^{54,55} Namely, above the critical field H'_N , magnon excitations condensate at low temperatures. Quantitatively, one would expect the field-induced transition temperature T_N proportional to $(H' - H'_N)^\varphi$, where the crossover exponent φ depends on the dimensionality and the magnon dispersion relation.^{56,57} In the inset of Fig. 8-Bottom, we replot our data as $\ln T_N$ versus $\ln (H' - H'_N)$. By fitting the data to $T_N \propto (H' - H'_N)^\varphi$, we obtain $\varphi = \sim 0.54(6)$ and $H'_N = 7.7(4)$ T. The extracted φ is too small for either effective dimensionality ($\varphi = 2/3$ for 3D and 1 for

2D). Measurements at lower temperatures are necessary for more accurate determination of the critical exponent.”

Fig. 8. (Bottom) Temperature-magnetic field phase diagram of **NiBO** from specific heat (C') and magnetization (χ). The green dash line presents the fitting $T_N \propto (H' - H'_N)^\phi$. Inset: Plot of $\ln T_N$ vs $\ln (H' - H'_N)$.

Reviewer #3

- There are confusions between NiBO and NiBO-d8. It is not always very clear what has been done on NiBO and on NiBO-d8.

We thank the reviewer for pointing out this issue!

We have added either **NiBO** or **NiBO-d₈** to places where this was not indicated. These places include captions of Figs. 7, S9, S10, S11, S12, S13, S21, S22, and S23.

- The Structural information of NiBO-d8 at 100K is labelled as NiBO in table S1, or is it really NiBO? It seems to be the result of SCXRD on NiBO-d8. This point could be clarified.

The reviewer is correct that in Table S1 in SI, **NiBO** should be **NiBO-d₈**. We have made the corrections.

- If this is indeed NiBO-d8, comparing structural information on the deuterated analog at 100 K to structural information on the hydrogenated analogue at 295 K is not enough. The difference in ring conformation may be due to the deuteration and not to temperature. This can easily be tested by measuring the d8 compound at 295K and see if it is in the orthorhombic or monoclinic phase.

Reviewer 1 also raised an issue regarding the comparison of the two structures. We have made the following response:

We agree with the reviewer [1] and have deleted the sentence: "The β angle of the unit cell in the 100 K structure deviated from 90° to ~116° placing the crystal system of the 100 K structure into the monoclinic (*C2*) system while the 295 K structure is in the orthorhombic (*Immm*) system." The discussion on the two structures is now as follows (p. S-6):

"The reported crystal structures at 295 K and **NiBO-d₈** at 100(2) K are mostly similar with some major differences, such as solving the structure of **NiBO-d₈** in the monoclinic *C2* (No. 5) space group instead of the orthorhombic *Immm* (No. 71)."

Reviewer 3 is correct that the protio and deuterio bipyridine ligands in **NiBO** and **NiBO-*d*₈** have different torsion motions, and studies in the area on bi-aryl rings have been published [e.g., M.K. Dahlgren et al., *J. Chem. Inf. Model.* **53**, 1191 (2013); W. Wei et al., *J. Chem. Inf. Model.* **59**, 4764 (2019); S.M.G. Sanfeliciano et al., *PLOS ONE* 2018-3-14, N. Soltanzadeh et al., *Polyhedron* **28**, 1343 (2009)]. Such thermal motions at a higher temperature may lead to crystal disorder. The structure at 295K published in 1999 (cif file available at CCDC: Database Identifier, MAHPOA; Deposition No., 1209054; <https://www.ccdc.cam.ac.uk/structures/Search?Doi=10.1021%2Fic990243w&DatabaseToSearch=Published>) shows larger ellipsoids for C2 and C3 atoms as well as a greater mean-square displacement tensor, giving what appears to be a planar bipyridine ligand. Thus, the structure at 295 K should probably be refined as a disordered structure instead of the reported planar structure.

At lower 100 K used in the single-crystal diffraction of **NiBO-*d*₈**, the torsion of the bipyridine ligand is frozen, giving the non-planar bipyridine ligand and the structure in the *C*2 space group. In X-ray diffraction, H and D atoms, both with 1 electron, are essentially indistinguishable. Given that **NiBO** and **NiBO-*d*₈** show Haldane properties at temperature <100 K, the more precise current structure of **NiBO-*d*₈** at 100 K (which is the lowest available to us) is more relevant.

- On p. 11 it appears that there is a problem in the description of Fig. 6. I don't see as the author claims that "X decreases then increases" as a function of increasing magnetic field. It seems to be true if one looks at the background corrected data on Fig. S9, but not in Fig. 6.

The reviewer is correct. We have revised the sentence (p. 11, bottom) to the following:

"Note that at magnetic field (H) \leq 9 T, the magnetic susceptibility decreases with decreasing temperature. Above 9 T, an upturn occurs at low temperatures."

- Fig. S21 is labelled as Fig. 21.

We have made the correction. Thank you!

- P. 18: "the peak at 10.0 cm⁻¹ is determined to originate from the transition..." how were the assignment done? A quick explanation would probably suffice.

We have revised the sentence to the following:

"The peak at 10.0(1) cm⁻¹ is assigned to originate from the transition **A**, $|0,0\rangle \rightarrow |1,\pm 1\rangle$, based on the spin gap $E_g = 7.5(5)$ cm⁻¹ from susceptibility data which agrees with that from the QMC calculations."

- I may have missed a point but why is the INS data from Fig. S22 different from those of Fig. 10 at 5 K?

The reviewer is correct that we did not indicate that Fig. S22 is for **NiBO**, while Fig. 10 is for **NiBO-d₈**. We have added **NiBO** to the caption of Fig. S22.

In addition, since Fig. 10 for **NiBO-d₈** is focused on the ≤ 30 cm⁻¹ region with magnetic peaks, most of the phonon spectrum at 5 K is not included in our previous manuscript. We have now added the entire experimental INS at 5 K and DFT-calculated spectra (≤ 4000 cm⁻¹) in Fig. S22-Bottom along with those of the non-deuterated **NiBO** in Fig. S22-Top. Fig. S23 has been added to compare INS spectra of **NiBO** and **NiBO-d₈** at 5 K, as the current work is a rare opportunity to use INS, which does not have symmetry-based selection rules for phonon spectra, to compare the phonon properties of H-containing **NiBO** and **NiBO-d₈**.

DFT-calculated phonon symmetries and modes of **NiBO-d₈** have been added in Table S4 (p. S-30), so they may be compared with those of **NiBO** in Table S3 given earlier.

REVIEWERS' COMMENTS

Reviewer #1 (Remarks to the Author):

The authors have answered all technical questions and I agree with their conclusions.

They have made clear in the response that the potential for isolated monolayer Haldane materials is the most exciting development. This paper convincingly demonstrates that the bipy compound is a good Haldane material, like the bpe and bpa analogues are suggested to be.

I would hope that any follow up paper demonstrates the ability to produce monolayers of this material.

Reviewer #2 (Remarks to the Author):

In their resubmission, the authors have satisfactorily addressed the referee comments.

Reviewer #3 (Remarks to the Author):

I stand by my original assessment.

I do not believe this article, although it is good work nicely written up as a manuscript, qualifies for publication in Nature Communications due to lack of significant novelty.

In fact it seems that the authors responded to all the minor points I suggested but did not answer to the main issue, about novelty, that was simply left out from their answer.

I suppose this must mean that the authors agreed with the statement and could therefore not refute it...

RESPONSE TO REVIEWERS' COMMENTS

We thank the reviewers for the comments. The following point-by-point response to reviewers' comments. The response is in red color.

Reviewer #1

The authors have answered all technical questions and I agree with their conclusions.

They have made clear in the reponse that the potential for isolated monolayer Haldane materials is the most exciting development. This paper convincingly demonstrates that the bipyr compound is a good Haldane material, like the bpe and bpa analogues are suggested to be.

I would hope that any follow up paper demonstrates the ability to produce monolayers of this material.

We are planning the studies of the conversion of NiBO into monolayers to probe the Haldane properties of a single layer of NiBO.

Reviewer #2

In their resubmission, the authors have satisfactorily addressed the referee comments.

We thank the reviewer.

Reviewer #3

I stand by my original assesment.

I do not believe this article, although it is good work nicely written up as a manuscript, qualifies for publication in Nature Communications due to lack of significant novelty.

In fact it seems that the authors responded to all the minor points I suggested but did not answer to the main issue, about novelty, that was simply left out from their answer.

I suppose this must mean that the authors agreed with the statement and could therefore not refute it...

We thank Reviewer 3 for the time to review the manuscript and for recognizing our work on NiBO. The reviewer referred to Refs. 31-33 in the first review. Reviewer 3 is correct that these papers reported MOFs with the Haldane or a possible Haldane state. However, the Haldane states of these MOFs are solely based on DC magnetic susceptibility data. In fact, in Ref. 33 on $[\text{Ni}(\mu\text{-pip})(\mu\text{-ox})]_n$ (pip = piperazine) in 2004, the authors called the susceptibility measurements “preliminary” and stated in the abstract: “we see a significant deviation between the observed and calculated magnetic susceptibilities at the lowest experimental temperatures. This may be due either to the formation of a Haldane quantum antiferromagnetic ground state or to single-ion zero-field splitting effects.” To our knowledge, no other paper has been published to probe or confirm the Haldane properties of the MOFs in Refs. 31-33.

We cite Refs. 31-33 in the Introduction of the manuscript (with a similar description of the works in the first version of the manuscript):

“Haldane spin-1 chains based on MOFs with oxalate-based ligands, $[\text{Ni}(\mu\text{-bpa})(\mu\text{-ox})]_n$ (bpa = 1,2-bis(4-pyridyl)ethane),³¹ $[\text{Ni}(\mu\text{-bpe})(\mu\text{-ox})]_n$ (bpe = 1,2-di(4-pyridyl)ethylene),³¹ $[\text{Ni}(\mu\text{-en})(\mu\text{-ox})]_n$,³² and $[\text{Ni}(\mu\text{-pip})(\mu\text{-ox})]_n$ (pip = piperazine),³³ have been reported to show the presence of a finite spin gap through DC magnetic susceptibility measurements....From DC magnetic susceptibility data, García-Couceiro and coworkers found that the spin gap = 7.32 and 10.29 cm^{-1} for $[\text{Ni}(\mu\text{-bpa})(\mu\text{-ox})]_n$ and $[\text{Ni}(\mu\text{-bpe})(\mu\text{-ox})]_n$, respectively.³¹ Similarly, Keene and coworkers obtained spin gap = 11.5 cm^{-1} for $[\text{Ni}(\mu\text{-en})(\mu\text{-ox})]_n$ from DC susceptibility data.³²”

The novelty of our work here includes the following:

- (1) Haldane properties of NiBO are reported for the first time in the current work.
- (2) Comprehensive studies of NiBO by a variety of advanced techniques reported in the present manuscript convincingly show that NiBO is Haldane spin-1 chains in a planar MOF with spin gaps between the singlet ground and the triplet excited state that is split by ZFS (zero-field

splitting). The techniques include variable-temperature powder neutron diffraction (PND), magnetization measurements (DC magnetic susceptibility at different fields), high-field ESR (HFESR), inelastic neutron scattering (INS), and specific-heat measurements. In addition, we have performed quantum Monte Carlo (QMC) simulations of the spin-1 chains in NiBO, which show excellent agreement with the experimental DC magnetic susceptibility data.

(3) X-ray single-crystal diffraction of deuterated NiBO- d_8 at 100 K reveals a non-planar structure of the 4,4'-bpy- d_8 ligand, reducing inter-chain magnetic couplings, a key towards making a Haldane state in NiBO.

Two features distinguish oxalate-based Haldane MOFs such as NiBO:

(1) These spin-1 chains are neutral in contrast to the positively charged spin-1 chains in, e.g., NENP and NDMAZ.

(2) The MOF-based compounds exhibit stacked 2D planar structures with Haldane 1D chains linked by N-containing ligands. In contrast, cationic Haldane chains containing anions in NENP and NDMAZ make the solids non-planar. The 2D-MOFs may be exfoliated by overcoming the relatively weak Van der Waals force between the layers. Exfoliation of the 2D-MOFs can yield 2D nanosheets with interesting low-dimensional properties.